# DYNAACT: Large Language Model Reasoning with Dynamic Action Spaces

**Xueliang Zhao**♠★∗    **Wei Wu**★†    **Jian Guan**★    **Qintong Li**♠    **Lingpeng Kong**♠†

♠The University of Hong Kong    ★Ant Group

`{xlzhao,qtli,lpk}@cs.hku.hk`
`{wuwei19850318, jianguanthu}@gmail.com`

## Abstract

In modern sequential decision-making systems, the construction of an optimal candidate action space is critical to efficient inference. However, existing approaches either rely on manually defined action spaces that lack scalability or utilize unstructured spaces that render exhaustive search computationally prohibitive. In this paper, we propose a novel framework named DYNAACT for automatically constructing a compact action space to enhance sequential reasoning in complex problem-solving scenarios. Our method first estimates a proxy for the complete action space by extracting general sketches observed in a corpus covering diverse complex reasoning problems using large language models. We then formulate a submodular function that jointly evaluates candidate actions based on their utility to the current state and their diversity, and employ a greedy algorithm to select an optimal candidate set. Extensive experiments on six diverse standard benchmarks demonstrate that our approach significantly improves overall performance, while maintaining efficient inference without introducing substantial latency. The implementation is available at `https://github.com/zhaoxlpku/DynaAct`.

## 1 Introduction

Recent advances in complex reasoning with Large Language Models (LLMs) [Achiam et al., 2023, Jaech et al., 2024] have established a prevalent self-improvement paradigm: given a mass of problems paired with final answers, model developers first search for correct reasoning paths from a base model using test time scaling strategies [Snell et al., 2024], then improve it to internalize these patterns [Guo et al., 2025] through supervised fine-tuning [Guan et al., 2025] or reinforcement learning [Guo et al., 2025]. While LLMs have exhibited remarkable reasoning capabilities, current approaches to long-term reasoning in these models often suffer from fundamental limitations. On the one hand, some approaches explicitly define an action space and state space and structure reasoning hierarchically, where action selection and state prediction are carried out iteratively [Hao et al., 2023, Qi et al., 2024] along the reasoning path. However, because action spaces are often heuristically designed, the resulting actions tend to either be too specific to generalize across domains or too broad to effectively guide reasoning. On the other hand, in the absence of an explicit definition for action spaces, other approaches impose a specific format on generation and perform reasoning in an autoregressive manner [Lightman et al., 2023, Guo et al., 2025]. These approaches inherently search the entire natural language space for reasoning, thereby necessitating powerful base models.

We investigate LLM reasoning within the framework of Markov Decision Process (MDP) [Hao et al., 2023], where a reasoning trace consists of a series of actions and states. Instead of focusing on

---

∗ This work was done during an internship at Ant Group.
† Corresponding authors.

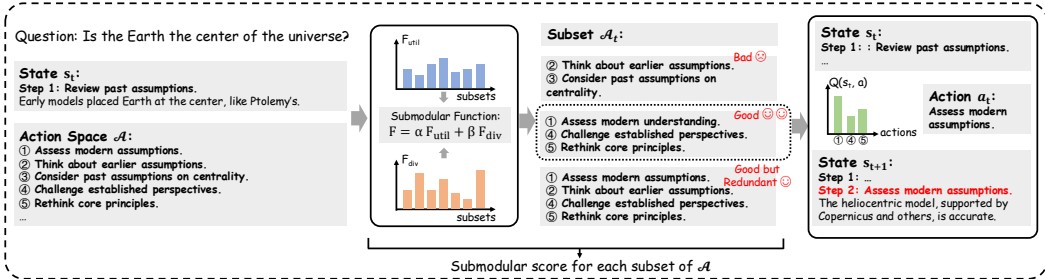

Figure 1: Overview of the proposed method. Given the proxy action space $\mathcal{A}$, the method searches for the subset $\mathcal{A}_t$ that maximizes the submodular function, which consists of a utility term and a diversity term. The subset $\mathcal{A}_t$ is then used for the subsequent reasoning steps.

policy learning or reward modeling, we pay special attention to action space construction, as a well-defined action space is fundamental to MDP-based reasoning. Specifically, we identify two essential properties that a qualified action space should possess: **(1) Scalability:** it should be automatically learned from demonstration data, rather than being manually engineered, to strike a balance between generalization and utility; and **(2) Compactness:** it should maintain a dynamically constructed, information-dense structure so that for each step desired action can be picked up from a small yet complete candidate set. The core technical challenge lies in developing a principled approach that simultaneously optimizes both objectives: distilling generalizable action patterns from demonstrations while eliminating redundant candidates that would otherwise impede efficient exploration.

To achieve both scalability and compactness, we frame the problem of action space construction as a subset selection task and propose DYNAACT, in which the action space for each reasoning step is dynamically determined by a submodular function that is learned through a data-driven approach. The key idea is to approximate a small subset from the entire action space that achieves the optimal balance between utility and diversity, leveraging the diminishing returns property of submodular functions to ensure linear computational complexity. Our method begins by extracting general reasoning patterns from a diverse corpus of complex problems to construct the complete action space. We then define a submodular function to evaluate candidate actions by jointly considering their utility to the current state and their diversity contribution. By maximizing this function via a greedy algorithm, we obtain an optimal subset of actions. Consequently, reasoning follows a standard Markov process: at each step, candidate actions are selected via the submodular optimization, an action is then chosen according to a Q-function estimated via Monte Carlo tree search, and a reasoning step is finally generated conditioned on the current reasoning context and the chosen action. Notably, throughout this process, only a lightweight embedding model used in the submodular function requires training, while the base LLM remains frozen.

We conduct extensive experiments on six benchmarks spanning general, reasoning, and math tasks. Evaluation results indicate that DYNAACT achieves significant improvements over baselines across all tasks, including MMLU, MMLU-Pro, GPQA, ARC-C, GSM8K, and MATH-500. Notably, DYNAACT excels at solving complex problems, achieving a 6.8% absolute gain over the recently proposed strong model rStar on MATH-500. Furthermore, an extended study demonstrates that while dynamic action space construction enhances efficacy, it does not introduce significant additional latency during inference compared to the baselines.

Our contributions are three-fold: (1) We propose dynamic action space construction as a novel research question, orthogonal to the extensive studies on LLM reasoning in the community; (2) We introduce a submodular function for action space construction, which significantly improves problem-solving accuracy while maintaining reasonable inference efficiency; and (3) We empirically verify the efficacy of our method across a wide range of tasks.

## 2 Preliminaries

Before delving into DYNAACT, we provide some background information. We begin by formulating MDP-based reasoning, then describe the search strategy applied. The section finally gives a brief introduction to submodular functions, which form the theoretical basis for action space construction.

## 2.1 Reasoning Framework

We formulate LLM reasoning as an MDP, defined by a tuple $(\mathcal{S}, \mathcal{A}, \mathcal{T}, \mathcal{R}, \gamma)$. Here, $\mathcal{S}$ represents the state space where each state $s_t \in \mathcal{S}$ encodes the cumulative reasoning context up to step $t$, with the initial state $s_0$ derived from the input prompt. $\mathcal{A}$ denotes the action space with $a_t \in \mathcal{A}$ an action indicating the progression of the reasoning context. The transition function $\mathcal{T} : \mathcal{S} \times \mathcal{A} \to \mathcal{S}$ determines how actions transform the reasoning state, while $\mathcal{R} : \mathcal{S} \times \mathcal{A} \to \mathbb{R}$ assigns rewards based on the quality of reasoning steps. The discount factor $\gamma \in [0, 1]$ balances immediate and future rewards.

At each time step $t$, the LLM selects an action $a_t$ from a candidate set $\mathcal{A}_t \subseteq \mathcal{A}$, which may be generated automatically [Hao et al., 2023] or specified manually [Qi et al., 2024]. In our implementation, the action selection is governed by a learnable value function $Q(s_t, a)$ that estimates the expected cumulative reward:

$$a_t = \arg\max_{a \in \mathcal{A}_t} Q(s_t, a),$$

$$Q(s_t, a) = \mathbb{E}\left[\sum_{k=0}^{\infty} \gamma^k \mathcal{R}(s_{t+k}, a_{t+k}) \mid s_t, a_t = a\right].$$

Our primary focus is on developing a principled approach to construct the candidate action set $\mathcal{A}_t$ at each step, as this significantly impacts the efficiency and efficacy of the reasoning process in complex problem-solving scenarios.

## 2.2 Monte Carlo Tree Search

We employ Monte Carlo Tree Search (MCTS) for estimating $Q(s_t, a)$ [Silver et al., 2016]. In a nutshell, the estimation is achieved by simulating multiple continuations of the reasoning process from the current state after applying the candidate action. During these simulations, the outcomes of the extended reasoning traces are evaluated, and the value of an action is approximated as the average result observed across the simulations. In this way, MCTS effectively balances the exploration of less frequently visited candidate actions with the exploitation of those that have demonstrated promising progress, which is critical to ensure that the estimated value function reliably reflects the potential benefit of each action in guiding the overall reasoning process. We defer additional technical details of MCTS, including how selection, expansion, simulation, and backpropagation are performed, to Appendix C.

## 2.3 Submodular Functions

The problem of action space construction can naturally be formalized as a subset selection task, where the goal is to identify a small, high-value subset from a much larger set of potential candidates. A typical approach to subset selection involves assessing the utility of each element, ensuring that every new element added contributes additional value to the overall set. To achieve this, submodular functions are often employed due to their diminishing returns property, which prioritizes the selection of elements that offer unique and informative value [Fujishige, 2005, Kothawade et al., 2022, Chen et al., 2024]. As a result, subset selection with submodular functions ensures that the marginal benefit of adding an element to a smaller subset is greater than adding it to a larger one, thus enhancing the compactness of the subset.

Formally, given two candidate sets $X \subseteq X'$ and an action $a \in X \setminus X'$, a submodular function $F(\cdot; \cdot)$ satisfies the following condition:

$$F(X \cup \{a\}; s_t) - F(X; s_t) \geq F(X' \cup \{a\}; s_t) - F(X'; s_t). \tag{1}$$

Heading toward a scalable approach for constructing compact action spaces, we take advantage of submodular functions. The problem then boils down to (1) how to define a proper submodular function; (2) how to learn the submodular function from data; and (3) how to perform action space construction with the submodular function, as will be presented in the following Section.

**Algorithm 1** Complete Pipeline of Sequential Reasoning.

---

**Require:** Input question $q$, dataset $\mathcal{D}$, number of groups $k$, candidate selection budget $m$, maximum reasoning steps $T$

1: /* Proxy Action Space Estimation (performed once) */
2: Partition the dataset $\mathcal{D}$ into $k$ groups: $\{\mathcal{D}_1, \mathcal{D}_2, \ldots, \mathcal{D}_k\}$
3: **for** $i = 1$ to $k$ **do**
4:      Extract an observation sketch $o_i = \langle a_1, a_2, \ldots, a_{|o_i|} \rangle \leftarrow \text{LLMQuery}(\mathcal{D}_i)$
5: Form the action space $\mathcal{A} = \bigcup_{i=1}^{k} o_i = \bigcup_{i=1}^{k} \langle a_1, a_2, \ldots, a_{|o_i|} \rangle$
6: Train the embedding function $\mathbf{e}$ using Q-learning objective (Eq. (5)) with observation sketches as demonstration data
7: $s_0 \leftarrow \text{InitializeState}(q)$
8: **for** $t = 0$ to $T - 1$ **do**
9:      /* Candidate Action Selection via Greedy Algorithm */
10:      $X \leftarrow \emptyset$
11:      **for** $i = 1$ to $m$ **do**
12:          $X_d \leftarrow \mathcal{A} \setminus X$
13:          $a^* \leftarrow \arg\max_{a \in X_d} F(X \cup \{a\}; s_t)$
14:          $X \leftarrow X \cup \{a^*\}$
15:      $\mathcal{A}_t(s_t) \leftarrow X$
16:      /* Action Evaluation using MCTS */
17:      **for all** $a \in \mathcal{A}_t(s_t)$ **do**
18:          $Q(s_t, a) \leftarrow \text{MCTS}(s_t, a)$
19:      /* Action Selection and State Update */
20:      $a_t \leftarrow \arg\max_{a \in \mathcal{A}_t(s_t)} Q(s_{t-1}, a)$
21:      $s_{t+1} \leftarrow \text{UpdateState}(s_t, a_t)$
**Output:** $\{s_0, a_0, s_1, \ldots, s_T\}$

---

## 3 Method

We detail the approach to constructing $\mathcal{A}_t$ given the current state $s_t$. Specifically, our method consists of three stages: estimating an approximation of the complete action space as $\mathcal{A}$ (§3.1), defining a submodular function $F(\mathcal{A}_t, s_t)$ based on $s_t$ (§3.2), and utilizing the function to derive $\mathcal{A}_t$ (§3.3). Figure 1 provides an overview of the method.

### 3.1 Proxy Action Space Estimation

We first estimate an approximation as a proxy of the complete action space, denoted as $\mathcal{A}$. Specifically, we follow Wang et al. [2024a] by employing observations as candidate actions, where observations are typically cues that guide the reasoning process (cf. Figure 1). Given a problem corpus (e.g., mathematical questions, logical puzzles, etc.), we randomly divide it into $k$ groups and feed each group to an LLM for observation collection. The division strategy ensures that each group is appropriately sized, avoiding prohibitive computational costs from the LLM. We then query the LLM to extract general observation sketchs per group that can be applied broadly and focus solely on the core operations. After that, the resulting observations are collected and form the proxy action space $\mathcal{A}$ with duplicate items removed. The prompt for observation sketch extraction is provided in Appendix D. Notably, $\mathcal{A}$ can be easily scaled up by incorporating examples from broader domains or fields; one can also develop various agents by applying the method to domain-specific corpora.

### 3.2 Submodular Function Definition

To construct an optimal candidate action set that balances both utility (in terms of expected rewards) and diversity, we propose a submodular function $F(\mathcal{A}_t; s_t)$ for a candidate subset $\mathcal{A}_t \subseteq \mathcal{A}$ as follows:

$$F(\mathcal{A}_t; s_t) = \alpha \, f_{\text{util}}(\mathcal{A}_t; s_t) + \beta \, f_{\text{div}}(\mathcal{A}_t), \tag{2}$$

where $f_{\text{util}}(\mathcal{A}_t; s_t)$ measures the expected utility of the candidate actions in advancing the reasoning process, $f_{\text{div}}(\mathcal{A}_t)$ promotes diversity within the selected set, and $\alpha, \beta$ are balancing parameters. To ensure that $F(\mathcal{A}_t; s_t)$ defined by Eq. (2) meets the condition given by Eq. (1), we define the utility

term as:

$$f_{\text{util}}(\mathcal{A}_t; s_t) = \log \left( \sum_{a \in \mathcal{A}_t} \exp \left( \mathbf{e}(s_t)^T \mathbf{e}(a) \right) \right), \tag{3}$$

where $\mathbf{e}(\cdot)$ is an embedding function that maps states and actions to a shared representation space. Then the diversity term is defined as

$$f_{\text{div}}(\mathcal{A}_t) = \sum_{a_i \in \mathcal{A}_t} \min_{\substack{a_j \in \mathcal{A}_t \\ a_j \neq a_i}} \left( 1 - \mathbf{e}(a_i)^T \mathbf{e}(a_j) \right), \tag{4}$$

This formulation encourages the selection of actions that are maximally distinct from each other in the embedding space, preventing redundancy in the candidate set.

**Lemma 1.** *Given the definitions of the relevance term $f_{\text{util}}(\mathcal{A}_t; s_t)$ in Eq. (3) and the diversity term $f_{\text{div}}(\mathcal{A}_t)$ in Eq. (4), the function $F(\mathcal{A}_t; s_t)$ defined in Eq. (2) is submodular with respect to the candidate action set $\mathcal{A}_t \subseteq \mathcal{A}$.*

The proof of Lemma 1 is provided in Appendix E.

A fundamental requirement of our framework is ensuring that $\mathcal{A}_t$ contains actions that maximize expected rewards in the reasoning process. To this end, we design the embedding function $\mathbf{e}(\cdot)$ to capture the effectiveness of actions in advancing the reasoning process. We formalize this requirement through Q-learning, where $\mathbf{e}(s_t)^T \mathbf{e}(a)$ approximates the Q-value—the expected future reward of executing action $a$ in state $s_t$. By incorporating this formulation into the standard Q-learning update equation [Watkins and Dayan, 1992, Reddy et al., 2019], we derive the following optimization objective:

$$\mathcal{L}(s_t, a, s_{t+1}) = \mathbf{e}(s_t)^T \mathbf{e}(a) - \left( r + \log \left( \sum_{a' \in \mathcal{A}} \exp \left( \mathbf{e}(s_{t+1})^T \mathbf{e}(a') \right) \right) \right)^2, \tag{5}$$

where the training data consists of state-action pairs $(s_t, a_t)$, with $a_t \in \mathcal{A}$ being the ground-truth action obtained from the observation sketches. The reward $r$ is defined as $r = 1$ for $a = a_t$ and $r = 0$ for all other actions $a \in \mathcal{A} \setminus \{a_t\}$. We train the embedding model by minimizing $\mathcal{L}$ over all states $s_t$ in the training set and all actions $a \in \mathcal{A}$. This objective ensures that the embedding function learns to prioritize actions that contribute substantively to the problem-solving progression.

### 3.3 Action Space Construction

With $\mathcal{A}$ and $F(\cdot; \cdot)$ at hand, we aim to derive $\mathcal{A}_t$ by selecting $m$ elements from $\mathcal{A}$ that maximize $F(\cdot; s_t)$. Formally, we construct $\mathcal{A}_t$ by solving

$$\mathcal{A}_t(s_t) = \arg \max_{X \subseteq \mathcal{A}, \, |X|=m} F(X; s_t). \tag{6}$$

Owing to the submodularity of $F(\cdot; \cdot)$, the combinatorial optimization problem (Eq. (6)) can be effectively approximated by a greedy algorithm [Nemhauser et al., 1978] with complexity of $O(m^2|\mathcal{A}|)$, where $|\mathcal{A}|$ denotes the size of $\mathcal{A}$. In practice, we begin with an empty set $X$ and iteratively add the element from $\mathcal{A} \setminus X$ that produces the highest marginal increase in $F(\cdot; \cdot)$ when combined with the current set $X$. Formally, for each iteration, we define the set of remaining candidates $X_d = \mathcal{A} \setminus X$ and select the element $a$ such that

$$a = \arg \max_{a \in X_d} F(X \cup \{a\}; s_t).$$

The process is repeated until $m$ elements have been selected, and the final set $X$ is then taken as $\mathcal{A}_t$.

Algorithm 1 summarizes the complete pipeline of our reasoning method, encompassing both action space construction and MCTS-based reasoning path search.

## 4 Experiments

### 4.1 Benchmarks

We employ six standard benchmarks covering three domains: general, reasoning, and math. Specifically, we use the following datasets:

(1) **MMLU** [Hendrycks et al., 2020] is a benchmark designed to evaluate a model's ability to answer a wide variety of tasks, including reading comprehension, reasoning, and problem-solving, across general domains. It is widely used for assessing language model performance in broad tasks; (2) **MMLU-Pro** [Wang et al., 2024b] is an extension of MMLU, containing more challenging and professional-level problems. This dataset tests the model's capabilities on more complex problems in general domains; (3) **GPQA** [Rein et al., 2023] focuses on evaluating reasoning and problem-solving skills, providing real-world open-domain problems that require advanced reasoning to solve; (4) **ARC-challenge (ARC-C)** [Clark et al., 2018] is part of the AI2 Reasoning Challenge and contains science-based multiple-choice questions. These questions require deep reasoning and are specifically designed to challenge models in complex reasoning tasks; (5) **GSM8K** [Cobbe et al., 2021] is a dataset consisting of grade-school level math word problems that require logical reasoning. It tests a model's ability to solve elementary-level math problems; and (6) **MATH-500** [Lightman et al., 2023] is a dataset containing high school-level math problems. It serves to assess a model's ability to handle more advanced mathematical reasoning.

## 4.2 Evaluation Metrics

We use exact match accuracy as the primary metric for evaluating the performance of our method. Specifically, for multiple-choice question-answering tasks, such as MMLU, MMLU-Pro, GPQA, and ARC-C, accuracy is calculated based on the exact match between the predicted choice and the ground-truth choice (typically denoted by a letter representing the correct answer). For math problem-solving tasks, such as GSM8K and MATH-500, accuracy is calculated by comparing the predicted final answer, enclosed by \boxed{}, with the ground-truth answer.

## 4.3 Baseline Methods

We compare the proposed method with the following baselines: (1) **Zero-shot CoT**: This baseline uses Llama 3.1 [Dubey et al., 2024] with zero-shot Chain-of-Thought (CoT) prompting [Wei et al., 2022], generating reasoning paths in a single pass; (2) **SC@maj16**: This method applies the self-consistency (SC) technique [Wang et al., 2022], where multiple reasoning paths are generated and the most frequent result is selected. We use the SC@maj16 variant, which runs 16 rollouts to increase the accuracy of reasoning; (3) **RAP**: The method [Hao et al., 2023] integrates a world model and a reasoning agent, balancing exploration and exploitation to efficiently find high-reward reasoning paths via MCTS. The action space is formed by automatically generated sub-questions; and (4) **rStar**: The method [Qi et al., 2024] utilizes 5 manually defined actions as the action space. Reasoning traces are searched with MCTS rollouts, and the final trace is determined by a small LLM as a discriminator.

All of these baselines use Llama-3.1-8B-Instruct [Dubey et al., 2024] as the backbone, consistent with our method. Additionally, both RAP and rStar use 16 rollouts, as in our method.

## 4.4 Implementation Details

For the proxy action space estimation (§3.1), we use the Open-Platypus [Lee et al., 2023] corpus, which covers a wide range of topics, including math, scientific reasoning, and more. The corpus contains a total of $24, 652$ problems, which are used to form the problem set. We divide the corpus into $k = 2, 500$ groups. To extract observations, we query Llama-3.1-70B-Instruct [Dubey et al., 2024], resulting in $40, 822$ observations in total. In our submodular function definition (§3.2), we set the balancing parameters $\alpha = 0.9$ and $\beta = 0.1$ to ensure a proper balance between utility and diversity. For embedding efficiency, we use Llama-3.2-1B-Instruct [Dubey et al., 2024] as the backbone and select the last token's embedding as the output of the embedding function $e(\cdot)$. The embedding function is fine-tuned using the Q-learning objective, with a total of $83, 083$ state-action pairs, and the learning rate is set to $1e - 5$. For the action space construction (§3.3), we set the size of the candidate action set $\mathcal{A}_t$ at each time step $m = 5$. We use Llama-3.1-8B-Instruct [Dubey et al., 2024] as the world model [Hao et al., 2023], which generates reasoning steps during the MCTS process.

## 4.5 Main Results

The results of our experiments, presented in Table 1, reveal several key observations: (1) Our method outperforms all baseline methods across the six evaluated benchmarks, achieving significant

Table 1: Evaluation results on different benchmarks. Numbers in bold denote the best performance.

| Model | General | | Reasoning | | Math | |
|---|---|---|---|---|---|---|
| | MMLU | MMLU-Pro | GPQA | ARC-C | GSM8K | MATH-500 |
| Zero-shot CoT | 68.87 | 43.45 | 31.82 | 81.06 | 76.12 | 45.40 |
| SC@maj16 | 69.66 | 49.36 | 34.34 | 80.63 | 86.66 | 52.00 |
| RAP | 69.46 | 48.70 | 38.89 | 85.41 | 87.79 | 51.60 |
| rStar | 68.61 | 48.81 | 36.87 | 86.43 | 87.11 | 54.20 |
| DYNAACT | **70.22** | **51.40** | **39.39** | **88.31** | **89.16** | **61.00** |

improvements in general, reasoning, and math tasks. This confirms the effectiveness of our method in a wide range of problem-solving tasks; (2) In the math domain, we observe the most notable improvements. Our method achieves 1.37% and 6.80% improvements over the baselines on GSM8K and MATH-500, respectively. The MATH-500 dataset, which requires a higher level of reasoning capability, particularly benefits from the use of the submodular function. The ability to construct a more compact and utility-optimized action space enables more efficient exploration, leading to improved performance in complex mathematical reasoning tasks; and (3) While rStar performs better than RAP on MATH-500, its performance in other benchmarks suffers due to the limited scalability of its manually defined action space. This limits rStar's ability to effectively handle the full range of tasks, making it less scalable compared to our method.

## 5    Discussions

In addition to the comprehensive evaluation across multiple benchmarks, we aim to dive deeper into DYNAACT to gain further insights into its underlying mechanisms. Specifically, we investigate the following research questions: (1) **RQ1:** How do different components affect performance? (2) **RQ2:** Can DYNAACT learn to have a compact action space, thereby facilitating efficiency in reasoning? (3) **RQ3:** What is the utility of the actions selected by DYNAACT? (4) **RQ4:** Does DYNAACT introduce additional latency during inference? Besides, we are also curious about (5) **RQ5:** How does reasoning performance vary across different levels of difficulty? and (6) **RQ6:** Can the submodular function enhance diversity of actions? We leave the discussions to **RQ5** and **RQ6** to Appendix F.

### 5.1    Ablation Study for RQ1

We exploit ARC-C and MATH-500 to strike a balance between difficulty and domain diversity, and examine four variants of DYNAACT, including: exclusion of the utility term, denoted as "- util"; exclusion of the diversity term, referred to as "- div"; submodular function without Q-learning defined by Eq. (5), denoted as "- q-learning", which uses Llama-3.2-1B-Instruct for embedding directly; and removal of the submodular function, represented as "- submodular",

Table 2: Ablation study.

| Model | ARC-C | MATH-500 |
|---|---|---|
| DYNAACT (full) | **88.31** | **61.00** |
| - util | 87.63 | 53.40 |
| - div | 86.52 | 53.80 |
| - q-learning | 87.80 | 55.80 |
| - submodular | 85.15 | 52.00 |

which generates action spaces using Llama-3.1-8B-Instruct directly.

The results shown in Table 2 indicate that the full version of DYNAACT achieves the best performance across both benchmarks, underscoring the critical importance of each component in our method. Removing the utility term leads to a slight decrease in performance, highlighting its contribution to overall reasoning effectiveness. Excluding the diversity term results in further performance degradation, emphasizing the need for diversity in the candidate action set. Omission of the Q-learning objective causes a noticeable drop in performance, demonstrating the necessity of this learning procedure for constructing an effective submodular function. Finally, the model without the submodular function performs the worst, reinforcing the essential role of the submodular strategy in achieving effective action selection.

## 5.2 Compactness Study for RQ2

We examine whether DYNAACT can produce compact action spaces. Ideally, a compact action space would enhance search efficiency, leading to better reasoning performance with a smaller size. Figure 2 compares DYNAACT with RAP where the actions are constrained to be sub-questions. From the results, we observe that: (1) The action spaces of RAP are highly redundant. Increasing the number of rollouts yields limited performance improvements when $m = 5$ or $m = 10$. Only when $m$ is increased to 15 do we observe noticeable performance gains with respect to the number of rollouts, though the marginal gains are still much slower compared to DYNAACT. (2) On the other hand, even with $m$ set to 5, DYNAACT still demonstrates significant performance improvements as more rollouts are carried out. The advantages over RAP remain consistent across all values of $m$, highlighting the efficacy of DYNAACT in generating compact action spaces.

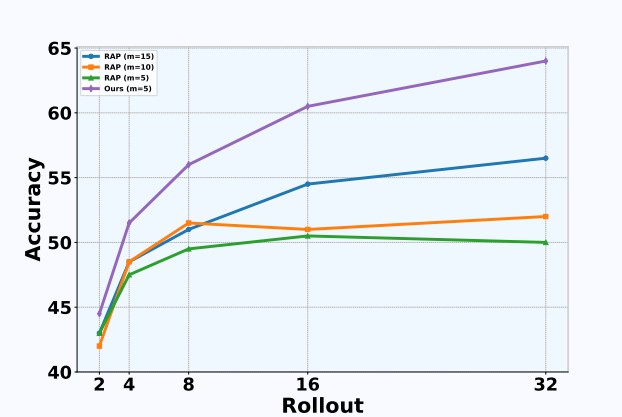

Figure 2: A comparison of RAP and DYNAACT with respect to different action space sizes (i.e., $m$). The $x$-axis indicates the number of rollouts, while the $y$-axis shows the accuracy on MATH-500.

## 5.3 Utility Study for RQ3

We select rStar as the baseline since its manually designed actions are presumed to be overly broad, limiting their utility for reasoning. Typically, there is no standard method for measuring "utility". Therefore, following Zhao et al. [2024], we identify critical steps in problem-solving (e.g., key insights, decisions, or calculations essential for solving a problem) and examine whether the reasoning steps triggered by actions contain these crit-

Table 3: Evaluation results on the utility of action spaces. "Accuracy" is measured on a subset of MATH-500 with Level 5 difficulty (i.e., the most difficult subset). This subset is chosen because the "utility" of actions (i.e., if critical steps are triggered) plays a more critical role in solving complex problems.

| Model | Critical Step Coverage (↑) | Accuracy (↑) |
|---|---|---|
| DYNAACT | 0.63 | 31.34 |
| rStar | 0.47 | 26.87 |

ical steps. The ratio of solutions containing critical steps (referred to as the *Critical Step Coverage*) is then used as a proxy metric for utility. Table 3 presents the evaluation results. We can see that DYNAACT enables more effective identification and triggering of essential reasoning steps, which also explains its superiority over rStar in terms of accuracy.

## 5.4 Latency Study for RQ4

To assess whether DYNAACT introduces additional latency during inference, we compare it with rStar and RAP based on the relative time required to complete tasks on the MATH-500 dataset. Specifically, relative time is calculated as the time taken by rStar or RAP, relative to the time taken by

Table 4: Comparison of relative time and accuracy for different methods.

| Method | Rel. Time (↓) | Accuracy (↑) |
|---|---|---|
| DYNAACT | 1.00 | 61.00 |
| rStar | 0.95 | 54.20 |
| RAP | 1.12 | 51.60 |

DYNAACT. We set the action space size
(denoted as $m$) to 5 and the number of rollouts to 16 for all methods. Table 4 presents the results, including accuracy on MATH-500 to examine the latency-accuracy trade-off. The results show that while DYNAACT incurs a slight increase in latency compared to rStar, it achieves substantial improvements in accuracy. When compared to RAP, our method demonstrates lower latency, which can be attributed to the fact that the most computationally intensive operation in constructing the action space $\mathcal{A}_t$ is the encoding of $\mathbf{e}(s_t)$, with $\mathbf{e}(a)$ precomputed and cached for subsequent use. This contrasts with RAP, where sub-questions are generated in real-time, leading to higher computational overhead. Additionally, the submodular function can be computed efficiently using an approximate algorithm with linear complexity, further contributing to the reduced latency of our approach.

## 6 Related Work

### 6.1 LLM Reasoning

The exploration of LLMs' reasoning capabilities has emerged alongside studies of prompting strategies, among which Wei et al. [2022] demonstrated that a simple prompt can elicit chain-of-thought reasoning in LLMs; Zhou et al. leveraged prompts to guide LLMs in breaking down complex problems into simpler ones; and Shinn et al. [2024] enabled LLMs to recognize their mistakes and perform self-correction. Soon after, the research on reasoning shifted focus from prompting to data curation [Yue et al., 2024, Wen et al., 2024] and learning methods [Zelikman et al., 2022, 2024], whereby the community witnessed rapid progress in tackling complex problems such as mathematics [Yu et al., 2023, Gou et al., 2023, Mitra et al., 2024, Toshniwal et al., 2024], coding [Luo et al., 2024], visual comprehension [Cheng et al., 2024, Hu et al., 2024], and decision-making [Chen et al., 2023]. Recently, the success of OpenAI o1 [Jaech et al., 2024] and DeepSeek r1 [Guo et al., 2025] has catalyzed the rise of test-time scaling [Snell et al., 2024], where increased computation at inference enables LLMs to engage in long-term reasoning and achieve significant improvements on Olympiad-level math [AIME-2024], challenging coding benchmarks [Jain et al., 2024], and graduate-level QA tasks [Rein et al., 2023]. Our study contributes to test-time scaling research but takes an orthogonal approach to most existing efforts. Rather than focusing on data curation [Muennighoff et al., 2025, Guan et al., 2025] or reinforcement learning [Guo et al., 2025, Qi et al., 2024], we aim to construct a compact yet effective action space to facilitate MDP-based reasoning.

### 6.2 Submodular Optimization

Submodular optimization [Fujishige, 2005] has been successfully used in many applications, including model interpretability [Elenberg et al., 2017, Chen et al., 2018, 2024] and computer vision [Pervez et al., 2023]. For instance, Elenberg et al. [2017] applied submodular functions to model interpretability, framing it as a combinatorial maximization problem for efficient model explanations. Similarly, Chen et al. [2018] used submodular functions for instance-wise feature selection to explain deep learning decisions. Pervez et al. [2023] introduced conditional Poisson sampling to select key features for image and text recognition, focusing on improving recognition accuracy. More recently, Chen et al. [2024] applied submodular subset selection to deep model attribution, enhancing interpretability by identifying critical regions and reducing misattribution. Our work applies submodular optimization to action space selection in sequential reasoning tasks. Unlike prior methods focused on feature selection and model attribution, our approach optimizes action selection for enhanced compactness and scalability, offering a novel method that improves performance in complex problem-solving.

## 7 Conclusion

We propose DYNAACT, a novel approach for automatically constructing compact action spaces to enhance sequential reasoning in complex problem-solving tasks. DYNAACT incorporates a submodular function that optimizes action selection based on both utility and diversity, leading to improved inference efficiency and performance. Extensive experiments across six standard benchmarks demonstrate that DYNAACT not only outperforms existing methods but also maintains efficient inference without introducing substantial latency. These results underscore the effectiveness and versatility of our approach in tackling a wide range of problem solving challenges.

## Acknowledgements

We extend our gratitude to the HKU NLP group and the anonymous reviewers for their invaluable suggestions, which significantly enhanced this work. This work was supported in part by the joint research scheme of the National Natural Science Foundation of China (NSFC) and the Research Grants Council (RGC) under grant number N_HKU714/21, and by the Ant Group Research Intern Program.

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

## A  Limitations

While DYNAACT represents a significant advancement in action space construction for reasoning tasks, there are several limitations that should be addressed in future work to fully realize its potential:

(1) DYNAACT demonstrates strong performance across general, reasoning, and math tasks. However, its reliance on MCTS for selecting specific actions from the constructed action space can be computationally intensive, especially for large or complex tasks. Alternative test-time scaling methods, such as beam search or other exploration strategies, could be explored to further optimize the balance between exploration and exploitation, improving efficiency without sacrificing reasoning accuracy. Expanding the model's ability to scale across different search algorithms remains a promising direction for future work.

(2) While DYNAACT improves reasoning, it still depends on pre-trained Llama models, which may face challenges when handling extremely large or highly specialized action spaces. Exploring methods to optimize the model's scalability, such as combining DYNAACT with a stronger backbone model, could help tackle this limitation in future research.

## B  Broader Impacts

This work proposes a framework for dynamic action space construction to improve sequential reasoning with language models. By enabling more structured, efficient, and interpretable decision-making, our approach has the potential to benefit applications such as educational tutoring systems, automated theorem proving, and scientific problem-solving—domains where reasoning efficiency and transparency are critical. The ability to distill compact, high-utility action spaces may also reduce reliance on brute-force search and large-scale inference, contributing to more sustainable and accessible AI systems.

At the same time, these benefits raise important concerns. Automatically constructed action spaces are learned from data and may reflect biases present in the underlying corpora, potentially reinforcing harmful patterns during reasoning. Furthermore, as our method enhances the ability of LLMs to perform multi-step reasoning, it may be misused to generate misleading arguments or automate complex forms of manipulation. While the explicit structure introduced by our framework offers interpretability benefits, future work should explore mechanisms for auditing, constraint enforcement, and safe deployment in sensitive domains.

## C  Details of the MCTS Process

In this appendix, we provide a detailed formal description of the Monte Carlo Tree Search (MCTS) procedure used to estimate the value function $Q(s, a)$ for candidate actions. The MCTS process comprises four stages: selection, expansion, simulation, and backpropagation. For simplicity, let the current state be denoted by $s$, and let the set of candidate actions from state $s$ be represented as $\mathcal{A}(s)$.

### Selection

Starting from the root node corresponding to the current state $s$, the selection phase traverses the tree by recursively choosing actions based on a balance between exploitation and exploration. Each node in the search tree is associated with two statistics:

$$N(s) \quad \text{(the number of visits to state } s\text{)}$$

and, for each action $a \in \mathcal{A}(s)$,

$$N(s, a) \quad \text{is the number of times that action } a \text{ has been selected from state } s,$$
$$Q(s, a) \quad \text{is the estimated average reward for taking action } a \text{ from state } s.$$

The action $a^*$ is selected at state $s$ according to the Upper Confidence Bound for Trees (UCT) formula:

$$a^* = \arg \max_{a \in \mathcal{A}(s)} \left[ Q(s, a) + c\sqrt{\frac{\ln N(s)}{N(s, a)}} \right],$$

where $c$ is an exploration constant. This selection continues recursively until a node is reached that is either a terminal state or not fully expanded (i.e., there exists an $a \in \mathcal{A}(s)$ such that the corresponding child node has not yet been created).

**Expansion**

Upon reaching a node $s$ that is not terminal and has unexplored actions, the expansion phase selects one such action $a' \in \mathcal{A}(s)$ for which no child node exists. A new child node $s'$ is then created to represent the state resulting from taking action $a'$ from $s$. The statistics for the new edge $(s, a')$ are initialized as:

$$N(s, a') = 0,$$
$$Q(s, a') = 0.$$

**Simulation**

From the newly expanded node $s'$, a simulation (or rollout) is conducted to estimate the value of the state-action pair $(s, a')$. The simulation proceeds by following a default policy—often a random or heuristic strategy—to generate a complete reasoning trajectory until a terminal state is reached. Let $r$ denote the reward obtained at the terminal state; this reward serves as an estimate for the value of taking action $a'$ from state $s$.

**Backpropagation**

Once the simulation concludes with reward $r$, the backpropagation phase updates the statistics along the path from the expanded node back to the root. For every state $s$ and action $a$ along this path, the updates are performed as follows:

$$N(s, a) \leftarrow N(s, a) + 1,$$
$$Q(s, a) \leftarrow Q(s, a) + \frac{r - Q(s, a)}{N(s, a)}.$$

Additionally, the visit count for each state along the path is updated:

$$N(s) \leftarrow N(s) + 1.$$

These updates refine the estimates $Q(s, a)$ based on the observed reward $r$, thereby improving the accuracy of the value function with successive simulations.

The MCTS process iterates through these steps—selection, expansion, simulation, and backpropagation—until a specified computational budget is reached.

# D  Prompt for Universal Problem-Solving Sketch Extraction

Below is the prompt used for extracting a universal problem-solving sketch from a set of problems. This prompt is designed to guide the extraction process so that the generated subgoals are broadly applicable, capture core actions, and can be flexibly applied across various problem domains.

This prompt ensures that the extracted subgoals capture the essential observations needed for constructing the proxy action space, and that they are effective for a wide range of problem-solving scenarios.

# E   Proof of Submodularity of $F(X; s_t)$

*Proof.* Let $V = \mathcal{A}$. A set function $f : 2^V \to \mathbb{R}$ is submodular if for every $X \subseteq Y \subseteq V$ and for every $x \in V \setminus Y$,

$$f(X \cup \{x\}) - f(X) \geq f(Y \cup \{x\}) - f(Y).$$

We write

$$F(X; s_t) = \alpha\, F_1(X) + \beta\, F_2(X),$$

where

$$F_1(X) = \log \left( \sum_{a \in X} \exp\left(\mathbf{e}(s_t)^T \mathbf{e}(a)\right) \right),$$

and

$$F_2(X) = \sum_{a \in X} m_X(a), \quad \text{with} \quad m_X(a) = \min_{b \in X \setminus \{a\}} \left(1 - \mathbf{e}(a)^T \mathbf{e}(b)\right).$$

**(i) Submodularity of $F_1$:** Define

$$g(X) = \sum_{a \in X} \exp\left(\mathbf{e}(s_t)^T \mathbf{e}(a)\right).$$

Note that for any $x \in V \setminus X$,

$$g(X \cup \{x\}) = g(X) + \exp\left(\mathbf{e}(s_t)^T \mathbf{e}(x)\right).$$

Thus, for $X \subseteq Y$ and $x \in V \setminus Y$, we have:

$$F_1(X \cup \{x\}) - F_1(X) = \log \left(1 + \frac{\exp\left(\mathbf{e}(s_t)^T \mathbf{e}(x)\right)}{g(X)}\right),$$

and

$$F_1(Y \cup \{x\}) - F_1(Y) = \log \left(1 + \frac{\exp\left(\mathbf{e}(s_t)^T \mathbf{e}(x)\right)}{g(Y)}\right).$$

Since $X \subseteq Y$ implies $g(X) \leq g(Y)$, it follows that

$$\frac{\exp\left(\mathbf{e}(s_t)^T \mathbf{e}(x)\right)}{g(X)} \geq \frac{\exp\left(\mathbf{e}(s_t)^T \mathbf{e}(x)\right)}{g(Y)},$$

and because $\log(1 + z)$ is an increasing function for $z \geq 0$, we obtain

$$F_1(X \cup \{x\}) - F_1(X) \geq F_1(Y \cup \{x\}) - F_1(Y).$$

Thus, $F_1$ is submodular.

**(ii) Submodularity of $F_2$:** For any $X \subseteq V$ and $x \in V \setminus X$, the marginal gain for $F_2$ is given by

$$\Delta_{F_2}(x \mid X) \triangleq F_2(X \cup \{x\}) - F_2(X).$$

For each $a \in X$, the value $m_X(a) = \min_{b \in X \setminus \{a\}} \left(1 - \mathbf{e}(a)^T \mathbf{e}(b)\right)$ updates upon addition of $x$ to

$$m_{X \cup \{x\}}(a) = \min \left\{ m_X(a), \left(1 - \mathbf{e}(a)^T \mathbf{e}(x)\right) \right\}.$$

Similarly, for the newly added element $x$,

$$m_{X \cup \{x\}}(x) = \min_{a \in X} \left(1 - \mathbf{e}(x)^T \mathbf{e}(a)\right).$$

Hence,

$$\Delta_{F_2}(x \mid X) = \sum_{a \in X} \left[ \min\{m_X(a), \left(1 - \mathbf{e}(a)^T \mathbf{e}(x)\right)\} - m_X(a) \right] + \min_{a \in X} \left(1 - \mathbf{e}(x)^T \mathbf{e}(a)\right).$$

Let us denote, for each $a \in X$,

$$\delta_X(a, x) = \min\{m_X(a), \left(1 - \mathbf{e}(a)^T \mathbf{e}(x)\right)\} - m_X(a).$$

Then,

$$\Delta_{F_2}(x \mid X) = \sum_{a \in X} \delta_X(a, x) + \min_{a \in X} \left(1 - \mathbf{e}(x)^T \mathbf{e}(a)\right).$$

Now, consider $X \subseteq Y \subseteq V$ and $x \in V \setminus Y$. For any $a \in X$, since $X \subseteq Y$ we have

$$m_Y(a) = \min \left\{ m_X(a), \min_{b \in Y \setminus X} \left(1 - \mathbf{e}(a)^T \mathbf{e}(b)\right) \right\} \leq m_X(a).$$

Thus,

$$\delta_Y(a, x) = \min\{m_Y(a), \left(1 - \mathbf{e}(a)^T \mathbf{e}(x)\right)\} - m_Y(a) \geq \min\{m_X(a), \left(1 - \mathbf{e}(a)^T \mathbf{e}(x)\right)\} - m_X(a) = \delta_X(a, x).$$

Also,

$$\min_{a \in Y} \left(1 - \mathbf{e}(x)^T \mathbf{e}(a)\right) \leq \min_{a \in X} \left(1 - \mathbf{e}(x)^T \mathbf{e}(a)\right).$$

Therefore,

$$\Delta_{F_2}(x \mid X) = \sum_{a \in X} \delta_X(a, x) + \min_{a \in X} \left(1 - \mathbf{e}(x)^T \mathbf{e}(a)\right),$$

$$\Delta_{F_2}(x \mid Y) = \sum_{a \in Y} \delta_Y(a, x) + \min_{a \in Y} \left(1 - \mathbf{e}(x)^T \mathbf{e}(a)\right).$$

Since $Y$ contains all elements of $X$ and possibly additional elements with non-positive marginal increments (i.e., $\delta_Y(a, x) \leq 0$ for $a \in Y \setminus X$), it follows that

$$\sum_{a \in X} \delta_X(a, x) \geq \sum_{a \in X} \delta_Y(a, x)$$

and

$$\min_{a \in X} \left(1 - \mathbf{e}(x)^T \mathbf{e}(a)\right) \geq \min_{a \in Y} \left(1 - \mathbf{e}(x)^T \mathbf{e}(a)\right).$$

Moreover, the additional terms from $Y \setminus X$ in $\Delta_{F_2}(x \mid Y)$ further decrease the total marginal gain. Thus, we obtain

$$\Delta_{F_2}(x \mid X) \geq \Delta_{F_2}(x \mid Y).$$

Hence, $F_2$ is submodular.

**(iii) Combination:** Since $F(X; s_t) = \alpha F_1(X) + \beta F_2(X)$ with $\alpha, \beta \geq 0$, it follows that for all $X \subseteq Y \subseteq V$ and for all $x \in V \setminus Y$:

$$F(X \cup \{x\}; s_t) - F(X; s_t) = \alpha \Delta F_1(x \mid X) + \beta \Delta F_2(x \mid X),$$

$$\geq \alpha \Delta F_1(x \mid Y) + \beta \Delta F_2(x \mid Y) = F(Y \cup \{x\}; s_t) - F(Y; s_t).$$

Thus, $F(X; s_t)$ is submodular. $\qquad\square$

Table 5: Evaluation results on the Level 3, Level 4, and Level 5 subsets of MATH-500. Numbers in bold denote the best performance.

|  | Level 3 | Level 4 | Level 5 |
|---|---|---|---|
| rStar | 72.38 | 50.78 | 15.67 |
| DYNAACT | **76.19** | **58.59** | **31.34** |
| - util | 68.57 | 52.34 | 17.16 |
| - q-learning | 71.43 | 53.13 | 20.90 |

# F    Additional Experimental Results and Analysis

## F.1    Reasoning Performance Across Difficulty for RQ5

We evaluate the effectiveness of both the utility term and the Q-learning objective using the MATH-500 dataset, which includes problems categorized by difficulty levels. This allows us to assess how each component impacts problem-solving performance across tasks of varying complexity. As shown in Table 5, the removal of the utility term leads to a more significant performance drop on harder problems (Level 5) compared to easier ones (Level 3). This indicates that the utility term is crucial for selecting actions that meaningfully contribute to the solution process, particularly for complex problems. In comparison, removing the Q-learning objective results in a slight performance drop. Without Q-learning, the embedding function $\mathbf{e}(\cdot)$ still selects actions relevant to the current state, but it fails to capture the long-term utility of those actions. As a result, while the model continues to choose relevant actions, the lack of Q-learning limits its ability to optimize the effectiveness of its actions over time.

We further compare our method with rStar, which demonstrates better performance than other baselines on MATH-500. However, due to rStar's manually defined action space, it faces challenges in scaling to more complex problems. As shown in the results for Level 5 problems in Table 5, rStar's performance drops significantly to 15.67%, whereas our method achieves a higher accuracy of 31.34%. This highlights the scalability advantage of our method.

## F.2    Action Diversity Analysis for RQ6

Table 6: Evaluation results showing the impact of the diversity term.

| Model | Diversity | Accuracy |
|---|---|---|
| Ours | 0.73 | 31.34 |
| - div | 0.49 | 24.63 |

To evaluate the impact of the diversity term, we follow Wang et al. [2024a] and calculate the diversity score of a candidate action set by measuring how dissimilar the actions are within the set. Specifically, the diversity score is determined by computing the ratio of dissimilar pairs of actions to the total number of possible action pairs in the set. This ratio is then averaged over all candidate action sets. We test on the Level 5 subset of the MATH-500 dataset, which consists of more complex problems that are sensitive to redundancy in the selected actions. The results are shown in Table 6. We observe that removing the diversity term results in a significant drop in the diversity score, which subsequently leads to a decrease in accuracy.

## F.3    Additional Analysis on Diversity Term in Submodular Function

To assess the sensitivity of DYNAACT to the choice of the diversity term $f_{\text{div}}$ in Eq. 4, we conducted additional experiments using two alternative diversity metrics: *Mean Pairwise Distance* and *Mean Cosine Distance*. These alternatives were chosen to evaluate whether simpler diversity formulations could yield comparable or improved performance.

As shown in Table 7, both alternative diversity metrics lead to a performance drop compared to our original formulation. We attribute this to the fact that these metrics do not preserve the submodular property, which is central to the efficiency and theoretical guarantees of our greedy subset selection algorithm.

Table 7: Performance comparison using different diversity terms.

| Method | Accuracy (MATH-500) |
|---|---|
| Mean Pairwise Distance | 57.80 |
| Mean Cosine Distance | 58.20 |
| DYNAACT | **61.00** |

## F.4 Resource Consumption Analysis

While §5.4 presents a relative latency comparison, we include here the raw inference time per example to offer a more complete view of resource consumption. All measurements were taken on an $8 \times$A100 GPU machine using the MATH-500 dataset, with each method evaluated under consistent rollout settings.

Table 8: Per-example raw inference time and accuracy on MATH-500 across methods.

| Method | Raw Time ($\downarrow$) | Accuracy ($\uparrow$) |
|---|---|---|
| Zero-shot CoT | 1.68s | 45.40 |
| SC@maj16 | 26.88s | 52.00 |
| RAP | 64.51s | 51.60 |
| rStar | 54.72s | 54.20 |
| DYNAACT | 57.60s | **61.00** |

As shown in Table 8, DYNAACT requires comparable runtime to other MCTS-based baselines such as RAP and rStar, while yielding significantly higher accuracy. Although MCTS-based approaches naturally incur more latency than single-pass generation methods like Zero-shot CoT, they also enable more effective reasoning. Since our method is orthogonal to the choice of search algorithm, future work could explore integration with more efficient test-time strategies (e.g., beam search or sample-efficient MCTS variants) to further reduce resource consumption while preserving accuracy.

## F.5 Comparison with Non-Submodular Selection Methods

To further understand the benefits of our submodular formulation, we compare DYNAACT against a RL-based pruning baseline. In this baseline, action space truncation is performed by selecting the top 5 candidate actions based solely on Q-value estimates, without considering submodular diversity or joint utility.

Table 9: Comparison of DYNAACT with an RL-based pruning baseline on the MATH-500 dataset.

| Method | Raw Time ($\downarrow$) | Accuracy ($\uparrow$) |
|---|---|---|
| RL-based Pruning | 56.89s | 53.20 |
| DYNAACT | 57.60s | **61.00** |

As shown in Table 9, while the RL-based pruning method achieves slightly lower inference latency, it significantly underperforms in accuracy compared to DYNAACT. This gap highlights the limitations of greedy RL-based pruning, which may select redundant or suboptimal actions. In contrast, our submodular approach explicitly optimizes for both utility and diversity, yielding a more compact yet expressive action set.

## F.6 Scalability with Large Proxy Action Spaces

To assess the scalability of DYNAACT to large-scale corpora, we varied the size of the proxy action space from $40,000$ to $1,000,000$ entries, while keeping the candidate set size $m = 5$ fixed. All experiments were conducted on the MATH-500 dataset.

As shown in Table 10, the inference latency increases moderately with larger proxy spaces—rising by approximately 18 seconds when scaling from 40k to 1M actions. Importantly, performance remains

Table 10: Scalability analysis of DYNAACT with varying proxy action space sizes on MATH-500.

| Proxy Action Space Size | Raw Time ($\downarrow$) | Accuracy ($\uparrow$) |
|---|---|---|
| 40k | 57.60s | 61.0 |
| 200k | 60.52s | 61.8 |
| 400k | 63.99s | 62.0 |
| 600k | 67.93s | 62.0 |
| 800k | 71.47s | 61.6 |
| 1M | 75.84s | 61.8 |

stable or slightly improves, indicating that DYNAACT effectively handles large candidate pools without significant degradation in efficiency. This scalability is largely attributed to the caching of action embeddings and the linear-time greedy algorithm used in submodular optimization.

## F.7 Empirical Study on the Utility-Diversity Trade-off

To better understand how the balancing parameters $\alpha$ and $\beta$ in Eq. 2 affect the trade-off between utility and diversity in the submodular function, we conducted a series of experiments on the MATH-500 dataset. We varied the relative weighting of the utility term $f_{\text{util}}$ and the diversity term $f_{\text{div}}$, while keeping their sum fixed ($\alpha + \beta = 1$).

| $\alpha, \beta$ | Accuracy (MATH-500) |
|---|---|
| (0.9, 0.1) | 61.00 |
| (0.7, 0.3) | 60.80 |
| (0.5, 0.5) | 55.40 |
| (0.3, 0.7) | 54.60 |
| (0.1, 0.9) | 54.80 |

Table 11: Effect of varying $\alpha$ and $\beta$ on model performance.

As shown in Table 11, performance is highly sensitive to the relative weighting of the utility term. Accuracy drops substantially when utility and diversity are given equal weight or when diversity dominates. However, once the utility coefficient $\alpha$ exceeds 0.7, the model achieves strong and stable performance. These findings suggest that while diversity contributes to a more robust action space, prioritizing utility is essential for effective reasoning in complex problem-solving tasks.

## F.8 Comparison with Few-Shot and Fine-Tuned Baselines

To assess whether the performance of DYNAACT could be attributed to exemplar-based prompting or supervised adaptation, we compare it with several baselines involving few-shot prompting and fine-tuning. The few-shot baseline retrieves the top 3 most similar examples from Open-Platypus using cosine similarity over embeddings from Llama-3.2-1B-Instruct, and provides them as context to Llama-3.1-8B-Instruct. The fine-tuned baseline directly trains Llama-3.1-8B-Instruct on the same corpus for 3 epochs with a learning rate of $1 \times 10^{-5}$. For both approaches, we also evaluate their self-consistency variants (SC@maj16), where 16 reasoning paths are sampled and majority voting is used for answer selection.

Table 12: Comparison of DYNAACT with few-shot and fine-tuned baselines across six benchmarks.

| Method | MMLU | MMLU-Pro | GPQA | ARC-C | GSM8K | MATH-500 |
|---|---|---|---|---|---|---|
| Zero-shot CoT | 68.87 | 43.45 | 31.82 | 81.06 | 76.12 | 45.40 |
| Few-shot baseline | 68.94 | 43.23 | 29.80 | 81.83 | 76.65 | 46.60 |
| Fine-tuned baseline | 68.72 | 43.56 | 34.34 | 82.08 | 78.17 | 48.20 |
| Few-shot SC@maj16 | 69.80 | 44.22 | 35.86 | 84.73 | 84.00 | 50.20 |
| Fine-tuned SC@maj16 | 69.89 | 45.89 | 34.85 | 86.18 | 84.38 | 52.60 |
| DYNAACT | **70.22** | **51.40** | **39.39** | **88.31** | **89.16** | **61.00** |

As shown in Table 12, both few-shot prompting and fine-tuning lead to modest improvements over the zero-shot baseline. However, DYNAACT consistently outperforms all baselines across tasks, demonstrating that its gains stem from principled, dynamic action space construction rather than static exemplar use or model adaptation alone.

## F.9 Case Study

To further demonstrate the effectiveness of DYNAACT, we present a series of case studies in Figures 3 - 5, each consisting of a "problem" and its corresponding predicted "solution". For each step in the solution, the process begins with the selection of an action (chosen using $Q(s_t, a)$, as described in §2.1), followed by the corresponding reasoning step. These examples demonstrate how DYNAACT enhances the effectiveness of the reasoning process.

---

**Problem:**
Compute: $1 - 2 + 3 - 4 + 5 - \cdots + 99 - 100$

**Solution:**
**Step 1:** I clarify the problem by identifying the key elements, constraints, and what I'm trying to find or achieve, and I rephrase it in my own words to ensure I understand it correctly.
We need to compute the sum of an alternating series where each term increases by 1, starting from 1 and ending at 100.

**Step 2:** I should look for connections, patterns, or relationships between the components, and consider how they interact or influence each other, whether it's in a mathematical, logical, or contextual sense.
The series alternates between addition and subtraction, which suggests a pattern where each pair of consecutive terms (one positive and one negative) could be combined to simplify the calculation.

**Step 3:** I'm searching for patterns, connections, and relationships between the different parts, and thinking about how they might help me solve the problem.
We can pair the terms as follows: $(1 - 2) + (3 - 4) + (5 - 6) + \cdots + (99 - 100)$. Each pair sums to $-1$.

**Step 4:** I evaluate my progress, checking if my approach is working and adjusting it as needed to ensure I'm on the right track.
There are 50 pairs in the series, each summing to $-1$.

**Step 5:** I break down complex problems into smaller, manageable parts, focusing on one aspect at a time, and identifying the essential variables, assumptions, and constraints that will help me make progress, which might involve creating a diagram, setting up equations, or developing a step-by-step plan.
The sum of the series is the sum of the 50 pairs, which is $50 \times (-1)$.

**Step 6:** Complete the solution and present the final answer within $\square$.

$$\boxed{-50}$$

---

Figure 3: Case study: solution to alternating series sum.

**Problem:**
The Greek army contained two types of soldiers: the upper class and the lower class soldiers. If there were a total of 5 upper class soldiers, and 10 lower class soldiers in a certain part of Athens, and the battle of Thermopylae demands a force of 4 upper class soldiers and 8 lower class soldiers, how many different battalions can be sent?

**Solution:**
**Step 1:** I clarify the problem by identifying the key elements, unknowns, and goals, and try to rephrase it in my own words to ensure I understand what's being asked.
We need to find out how many different groups (battalions) can be formed from a mix of 5 upper class soldiers and 10 lower class soldiers, where each group must consist of 4 upper class soldiers and 8 lower class soldiers.

**Step 2:** I look for opportunities to use analogies, metaphors, or abstractions to reframe the problem or make it more tractable, and I think about how I can leverage my existing knowledge or experience to inform my solution.
This problem can be approached by considering it as a combinatorics problem, specifically involving combinations since the order of selection does not matter.

**Step 3:** I consider the implications and potential consequences of my solution, and I think about how I can test, validate, or refine it further to ensure it's correct, complete, and relevant to the original problem.
To form a battalion, we need 4 upper class soldiers out of 5 and 8 lower class soldiers out of 10. The number of ways to choose 4 upper class soldiers from 5 is given by the combination formula $C(n,k) = \frac{n!}{k!(n-k)!}$, where n is the total number of items, and k is the number of items to choose. For the upper class, this becomes $C(5,4)$. Similarly, for the lower class, it's $C(10,8)$.

**Step 4:** I consider how the problem might be transformed or re-expressed in a way that makes it more tractable – can I reframe it in simpler terms, or find an analogous problem that's easier to solve, or identify a key insight that unlocks a more straightforward path to the solution?
The formula for combinations is $C(n,k) = \frac{n!}{k!(n-k)!}$. For the upper class, this is $C(5,4) = \frac{5!}{4!(5-4)!} = 5$. For the lower class, this is $C(10,8) = \frac{10!}{8!(10-8)!} = 45$.

**Step 5:** Are there any constraints, assumptions, or rules that I need to take into account when exploring solutions or possibilities?
Since the selection of upper class soldiers and lower class soldiers is independent of each other, the total number of different battalions that can be sent is the product of the number of ways to choose the upper class soldiers and the number of ways to choose the lower class soldiers.

**Step 6:** Once I think I have a solution, I'll consider its implications and potential limitations. I'll ask myself questions like 'What are the potential consequences of this solution?', 'Are there any potential biases or errors in my thinking?', and 'How might this solution impact different stakeholders or scenarios?'
The total number of battalions is the product of the number of ways to choose the upper class and the lower class soldiers, which is $5 \times 45 = 225$.

**Step 7:** Look for patterns, symmetries, and analogies that can help me understand the problem more deeply and make connections to other areas of knowledge or experience – are there any clever ways to reframe, reorganize, or reinterpret the problem that could reveal new insights or opportunities?
The solution is straightforward and based on the combination formula.

**Step 8:** Complete the solution and present the final answer within ☐.

$$\boxed{225}$$

Figure 4: Case study: solution to Greek army battalion formation problem.

**Problem:**
A hexagon is inscribed in a circle:

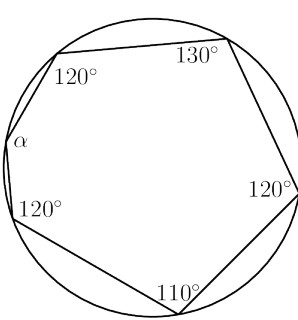

What is the measure of $\alpha$, in degrees?

**Solution:**
**Step 1:** I need to identify the underlying structure or pattern in the problem, whether it's a mathematical relationship, a logical argument, or a conceptual framework.
The problem involves an inscribed hexagon in a circle, with specific angles labeled. The goal is to find the measure of angle $\alpha$.

**Step 2:** I will decide on a suitable representation or model for the problem, and determine the best approach to explore the solution space, whether it's through calculation, simulation, or other means.
To find the measure of $\alpha$, I can use the fact that the sum of the interior angles of a hexagon is $720°$. Additionally, since the hexagon is inscribed in a circle, the measure of each central angle formed by two adjacent vertices of the hexagon is equal to the measure of the corresponding inscribed angle.

**Step 3:** I evaluate my progress, checking for errors, inconsistencies, and areas for improvement, and I adapt my approach as needed to ensure that I'm making progress toward a solution that is accurate, complete, and relevant.
Since the sum of the interior angles of a hexagon is $720°$, and the sum of the measures of the given angles is $105° + 110° = 215°$, the measure of angle $\alpha$ can be found by subtracting the sum of the given angles from the total sum of the interior angles of a hexagon.

**Step 4:** I break down complex problems into smaller, manageable parts, focusing on one aspect at a time, and identifying the essential variables, assumptions, and constraints that will help me make progress, which might involve creating a diagram, setting up equations, or developing a step-by-step plan.
Let's denote the measure of angle $\alpha$ as $x$. Since the sum of the interior angles of a hexagon is $720°$, we have the equation $x + 215° = 720°$.

**Step 5:** Are there any constraints, assumptions, or rules that I need to take into account when exploring solutions or possibilities?
The problem does not provide any additional constraints or assumptions beyond the given information.

**Step 6:** I should look for potential paths forward by asking myself questions like 'What are the implications of this?', 'How does this relate to that?', 'What would happen if I...', and 'What are the boundaries or constraints of this problem?' - this helps me generate possibilities and explore different avenues for solving the problem.
Now, let's solve the equation $x + 215° = 720°$ for $x$.

**Step 7:** I break down complex problems into smaller, manageable parts, focusing on one aspect at a time, and identifying the essential variables, assumptions, and constraints that will help me make progress, which might involve creating a diagram, setting up equations, or developing a step-by-step plan.
To solve for $x$, we can subtract $215°$ from both sides of the equation: $x = 720° - 215°$.

**Step 8:** Complete the solution and present the final answer within ▢.
$x = 720° - 215° = 505°$. However, since the measure of an angle cannot exceed $180°$, we need to subtract $360°$ from $505°$ to get the correct measure of angle $\alpha$. Therefore, $\alpha = 505° - 360° = 145°$.

$$\boxed{145}$$

Figure 5: Case study: solution to inscribed hexagon angle problem.

