# OpenReview forum: "DynaAct: Large Language Model Reasoning with Dynamic Action Spaces"
_NeurIPS.cc/2025/Conference — NeurIPS 2025 poster_

### Official Review · Reviewer_HY5o · 2025-06-30

**Clarity:** 3
**Significance:** 2
**Originality:** 3
**Rating:** 4
**Confidence:** 5

**Summary:**

This paper proposes DynaAct, a framework for dynamically constructing compact action spaces to enhance sequential reasoning in large language models (LLMs). The framework first extracts a proxy action space from demonstration data, and defines a submodular function to balance utility and diversity. Then, it uses greedy optimization to select optimal action subsets at each reasoning step. While the proposed approach shows improvements on six benchmarks, it suffers from fundamental methodological flaws and cost-effectiveness issue that severely limit its practical value.

**Questions:**

1. Whether the authors considered potential duplication caused by semantic similarity among the 40,822 actions used for selecting high-level strategies. Did they filter out different surface forms corresponding to the same strategy?
2. Is the "action" in MCTS referred to as the whole reasoning process or a step of the process?
3.  I would like to know more about the effect of the choice of m on the experimental results.
4. I would like to see some examples on the value of submodular function, is the function value consistent with human intuition?

**Ethical Concerns:**

["NO or VERY MINOR ethics concerns only"]

**Final Justification:**

I have raised the clarity score because the authors have clarified most of my concerns. And I keep my overall rating since it is positive.

**Limitations:**

yes

**Quality:**

3

**Strengths And Weaknesses:**

Strengths:
1. Dynamic action space construction is a relevant challenge in LLM reasoning, offering a middle ground between rigid manually-defined actions and unconstrained generation.
2. The evaluations span multiple domains (general, reasoning, math) across six benchmarks, which demonstrate the proposed framework's advantage.
3. The paper is clearly written.

Weaknesses:
1. The paper focuses on single-step math and reasoning tasks. However, such problems do not have a clearly defined notion of “step”. Modeling steps as an MDP process, as described in the preliminary section, may not be appropriate. Step-level reasoning is more suitable for agentic tasks such as WebArena [1] and ScienceWorld [2].
2. The authors employ a small model to select high-level strategies that guide an 8B frozen model in reasoning tasks. The authors primarily compare this approach with actor-critic baselines. However, I wonder whether this architecture can outperform training-based methods. Training-based approaches can significantly reduce inference time compared to the proposed method, which requires approximately 60 seconds per problem.
3. Generalization remains a concern. Can the strategies summarised from the current six datasets and the trained 1B critic generalize to OOD datasets such as AIME 2024/2025?

[1] WebArena: A Realistic Web Environment for Building Autonomous Agents
[2] ScienceWorld: Is your Agent Smarter than a 5th Grader?

---

> ### Author Rebuttal · Authors · 2025-07-30
>
> We genuinely appreciate your considerate and thorough feedback. We thoroughly respond to all of your concerns and questions below. We will also make sure to incorporate your insightful suggestions into the final version of the paper.
>
> ---
>
> ### **Q1: On the applicability of step-level MDP modeling**
>
> You raised a valid point regarding the notion of “step” in math and reasoning tasks, and the appropriateness of modeling them as MDPs. To clarify, our framework does not focus on single-step problems. In our setting, each “action” in the MCTS corresponds to **a single reasoning step** rather than the entire reasoning trace.
>
> As you pointed out, agentic tasks such as *WebArena* and *ScienceWorld* are indeed more naturally framed as multi-step MDPs. We fully agree, and to further assess the broader applicability of our method beyond standard reasoning tasks, we applied DynaAct to both benchmarks. For evaluation, we adopt the official metrics defined by each benchmark and report the average performance over 10 independent runs to mitigate statistical variance. The results are as follows:
>
> | Model | ScienceWorld | WebArena |
> | --- | --- | --- |
> | LLaMA-3.1-8B-Instruct | 35.6 | 6.5 |
> | SC@maj16 | 43.6 | 10.0 |
> | RAP | 45.9 | 11.9 |
> | rStar | 44.2 | 10.9 |
> | **DynaAct** | **56.5** | **17.4** |
>
> Our framework exhibits a considerable performance advantage over all baseline models, showcasing its versatility even in more intricate, agent-like settings.
>
> ---
>
> ### **Q2: Comparison with training-based methods**
>
> We appreciate this concern and would like to clarify that many training-based methods require significant **data curation and fine-tuning effort**, which limits their scalability and transferability across tasks. Moreover, in the context of strong instruct models (like Llama-3.1-8B-Instruct), we observed **only marginal improvements** with training-based methods.
>
> To illustrate this, we compared our method with a training-based variant, where Llama-3.1-8B-Instruct was fine-tuned on the training sets of GSM8K and MATH-500, and evaluated on their respective test sets:
>
> | Method | GSM8K | MATH-500 |
> | --- | --- | --- |
> | Llama-3.1-8B-Instruct | 76.1 | 45.4 |
> | Training-based | 78.5 | 48.2 |
> | **DynaAct (Ours)** | **89.2** | **61.0** |
>
> Furthermore, for certain benchmarks such as GPQA and MMLU, no training data is available, making our plug-and-play design particularly valuable.
>
> ---
>
> ### **Q3: Generalization to OOD tasks (e.g., AIME)**
>
> This is an excellent question that aligns with our core motivation. To evaluate this, we tested DynaAct on both AIME 2024 and AIME 2025 using **Qwen3** as the backbone (due to Llama-3.1’s low base performance on these challenging tasks). The results are:
>
> ### **Qwen3-1.7B**
>
> | Method | AIME 2024 | AIME 2025 |
> | --- | --- | --- |
> | Qwen3-1.7B | 13.4 | 9.8 |
> | SC@maj16 | 23.3 | 13.3 |
> | RAP | 23.3 | 16.7 |
> | rStar | 20.0 | 13.3 |
> | **DynaAct** | **30.0** | **23.3** |
>
> ### **Qwen3-4B**
>
> | Method | AIME 2024 | AIME 2025 |
> | --- | --- | --- |
> | Qwen3-4B | 25.0 | 19.1 |
> | SC@maj16 | 33.3 | 20.0 |
> | RAP | 30.0 | 23.3 |
> | rStar | 30.0 | 20.0 |
> | **DynaAct** | **36.7** | **26.7** |
>
> ### **Qwen3-8B**
>
> | Method | AIME 2024 | AIME 2025 |
> | --- | --- | --- |
> | Qwen3-8B | 29.1 | 20.9 |
> | SC@maj16 | 33.3 | 20.0 |
> | RAP | 30.0 | 23.3 |
> | rStar | 36.7 | 23.3 |
> | **DynaAct** | **46.7** | **30.0** |
>
> ### **Qwen3-14B**
>
> | Method | AIME 2024 | AIME 2025 |
> | --- | --- | --- |
> | Qwen3-14B | 31.7 | 23.3 |
> | SC@maj16 | 36.7 | 23.3 |
> | RAP | 33.3 | 26.7 |
> | rStar | 36.7 | 20.0 |
> | **DynaAct** | **43.3** | **33.3** |
>
> ### **Qwen3-32B**
>
> | Method | AIME 2024 | AIME 2025 |
> | --- | --- | --- |
> | Qwen3-32B | 31.0 | 20.2 |
> | SC@maj16 | 36.7 | 26.7 |
> | RAP | 33.3 | 23.3 |
> | rStar | 36.7 | 23.3 |
> | **DynaAct** | **46.7** | **30.0** |
>
> These results confirm that our method retain effectiveness on unseen, harder evaluations.
>
> ---
>
> ### **Q4: On semantic duplication in action space**
>
> Currently, we only filter actions based on surface-form duplicates (i.e., exact matches). We will make this point clearer in the final version.
>
> ---
>
> ### **Q5: Sensitivity to the choice of m**
>
> Thank you for suggesting a deeper analysis of the parameter $m$, which controls the size of the action space at each reasoning step.
>
> Here are the empirical results across six benchmarks:
>
> |  | MMLU | MMLUPro | GPQA | ARC-C | GSM8K | MATH-500 |
> | --- | --- | --- | --- | --- | --- | --- |
> | m = 3 | 69.3 | 50.1 | 37.9 | 88.1 | 86.3 | 57.6 |
> | m = 5 | 70.2 | 51.4 | 39.4 | 88.3 | 89.2 | 61.0 |
> | m = 7 | 70.3 | 51.9 | 38.9 | 89.1 | 87.3 | 60.2 |
> | m = 9 | 70.8 | 50.6 | 38.4 | 88.2 | 88.9 | 59.4 |
>
> As shown, $m = 5$ achieves a strong tradeoff between performance and redundancy. Beyond that, marginal gains taper off or slightly regress, possibly due to an overly large candidate set introducing distractors.
>
> ---
>
> ### **Q6: Human intuition vs. submodular function value**
>
> Thank you for your thoughtful question regarding the interpretability of our submodular objective. To demonstrate how the function behaves and whether it aligns with human intuition, we provide a concrete example below.
>
> Given the following reasoning context:
>
> > Question: Is the Earth the center of the universe?
> >
> >
> > Answer: *I clarify the problem by identifying the assumptions made in the past and restating them in my own words to better understand how they shaped the current situation. Early models placed Earth at the center, like Ptolemy’s.*
> >
>
> We trace the evolution of the action space during the submodular optimization process:
>
> |  | Action Space | Submodular Score |
> | --- | --- | --- |
> | step 1 | **action 1**: I need to identify the underlying structure or logic behind the current understanding, and examine how it fits into a broader conceptual or analytical framework; **action 2**: I look for opportunities to reinterpret the problem using analogies or alternative viewpoints, questioning traditional interpretations and seeing if other models from my experience could offer a better fit. | 0.61 |
> | step 2 | **action 1**: I need to identify the underlying structure or logic behind the current understanding, and examine how it fits into a broader conceptual or analytical framework; **action 2**: I look for opportunities to reinterpret the problem using analogies or alternative viewpoints, questioning traditional interpretations and seeing if other models from my experience could offer a better fit; **action 3** (new): I clarify the problem by re-examining its foundational principles and constraints. | 0.87 |
> | step 3 | **action 1**: I need to identify the underlying structure or logic behind the current understanding, and examine how it fits into a broader conceptual or analytical framework; **action 2**: I look for opportunities to reinterpret the problem using analogies or alternative viewpoints, questioning traditional interpretations and seeing if other models from my experience could offer a better fit; **action 3**: I clarify the problem by re-examining its foundational principles and constraints; **action 4** (new): I clarify the problem by identifying the initial assumptions that might have shaped the current understanding. | 0.89 |
>
> As you can see, from **step 1 to step 2**, the submodular score increases significantly, reflecting a meaningful gain in both utility and diversity. The newly added action (Action 3) introduces a distinct angle of reasoning, thereby enriching the action space.
>
> However, from **step 2 to step 3**, the increase becomes marginal. This is because Action 4 shares semantic overlap with Action 2 and Action 3 (all aiming at clarification and reframing), which leads to **diminishing marginal gains** under the submodular objective. This behavior aligns well with human intuition — once a range of complementary reasoning strategies is already covered, adding similar ones contributes less value.
>
> ---
>
> We truly appreciate the time and expertise you dedicated to reviewing our work. Your comments have helped sharpen the focus and rigor of our contributions. Please don’t hesitate to let us know if further clarifications are needed.

---

> ### Comment · Reviewer_HY5o · 2025-08-05
> **Response to author's rebuttal**
>
> Thanks a lot for the authors' response. I have raised the clarity score as the authors have clarified most of my concerns. And I keep my overall rating since it is positive.

---

> > ### Author Response · Authors · 2025-08-06
> > **Sincere thanks for your feedback!**
> >
> > Thank you for taking the time to go through our response. We're happy that we could clear up most of those concerns you had, and we appreciate you updating the clarity score.

---

### Official Review · Reviewer_ML8s · 2025-07-02

**Clarity:** 3
**Significance:** 3
**Originality:** 3
**Rating:** 5
**Confidence:** 3

**Summary:**

This paper identifies the limitation of the current framework on MDP-style LLM reasoning on lacking a scalable and compact action space. In response, the authors propose to first mine action space from existing corpora and formulate a submodular function to evaluate the utility and diversity of the actions to form a dynamic action space. Extensive experiments on six tasks show good performance improvement over previous methods with good latency control.

**Questions:**

1.	Will the authors open-source the observations extracted in Section 4.4 to facilitate future research?

**Ethical Concerns:**

["NO or VERY MINOR ethics concerns only"]

**Final Justification:**

The reviewer acknowledges the effort of the authors, which solved most of my concerns (backbones and other baselines).

**Limitations:**

yes

**Quality:**

3

**Strengths And Weaknesses:**

Strengths

1. This paper is relatively well-written and easy to follow.

2. The authors conduct experiments on a wide range of tasks, including both QA and math tasks.

3. The analysis is well-designed, including compactness, utility, and real-time latency.


Weakness

1. Studies on other backbones can further consolidate the performance gain over baselines and show the generalization of the learned submodular function on other models, e.g., Qwen 2.5, and other-sized models, e.g., 1.5B, 70B models. It is unclear how robust the action evaluation function is with other models or if it transfers among models.

2. It would be preferred if the authors could conduct qualitative analysis, e.g., examples of the actions, success and failure examples of the proposed methods, and change of action space at different steps of the optimization, etc.

3. Further comparison with few-shot prompting methods is needed since the dynamic action space selection is similar to dynamic exemplar selection [1] in terms of the extra prior introduced to the frozen inference models. Currently, in F.8 (page 19), the authors experiment with few-shot baselines with embeddings from Llama 3.2 1B is not the common practice. What is the performance of newer models such as GritLM [1], Gecko [2], or Instructor [3]?

[1] Learning To Retrieve Prompts for In-Context Learning: https://arxiv.org/pdf/2112.08633

[2] GritLM: https://arxiv.org/abs/2402.09906

[3] Gecko: https://arxiv.org/abs/2403.20327

[4] Instructor: https://arxiv.org/abs/2212.09741

---

> ### Author Rebuttal · Authors · 2025-07-30
>
> We sincerely thank you for your detailed and constructive review. We appreciate your positive comments on the writing quality, the breadth of the evaluation, and the analysis. Below, we carefully address each of your insightful concerns. We will also make sure to incorporate your insightful suggestions into the final version of the paper.
>
> ---
>
> ### **Q1: Generalization across model backbones and sizes**
>
> Thank you for raising this important point regarding the generalizability of the proposed method. We have conducted additional experiments using Qwen-2.5 models of various sizes (1.5B, 7B, and 72B) to evaluate the robustness and transferability of the submodular action selection framework. Below are the results:
>
> ### **Qwen-2.5-1.5B-Instruct**
>
> | Method | GSM8K | MATH-500 |
> | --- | --- | --- |
> | Qwen-2.5-1.5B-Instruct | 73.2 | 55.2 |
> | SC@maj16 | 82.4 | 60.4 |
> | RAP | 83.2 | 61.0 |
> | rStar | 81.6 | 62.6 |
> | **DynaAct** | **86.2** | **68.8** |
>
> ### **Qwen-2.5-7B-Instruct**
>
> | Method | GSM8K | MATH-500 |
> | --- | --- | --- |
> | Qwen-2.5-7B-Instruct | 91.6 | 75.8 |
> | SC@maj16 | 93.2 | 81.4 |
> | RAP | 92.1 | 81.2 |
> | rStar | 93.4 | 81.8 |
> | **DynaAct** | **95.7** | **84.0** |
>
> ### **Qwen-2.5-72B-Instruct**
>
> | Method | GSM8K | MATH-500 |
> | --- | --- | --- |
> | Qwen-2.5-72B-Instruct | 95.8 | 83.2 |
> | SC@maj16 | 95.4 | 86.6 |
> | RAP | 96.0 | 86.6 |
> | rStar | 95.5 | 85.4 |
> | **DynaAct** | **96.7** | **89.2** |
>
> These results demonstrate that DynaAct consistently outperforms strong baselines across different model sizes.
>
> ---
>
> ### **Q2: Qualitative analysis of action selection process**
>
> You bring up an important point regarding interpretability. While our primary focus has been on quantitative evaluation, we fully agree that qualitative insights are essential for understanding how and why the method behaves as it does.
>
> In addition to the three examples in Appendix F.9, we are glad to include more comprehensive failure cases to highlight the limitations and edge behaviors of the method in the final version. Furthermore, we provide a more detailed illustration of how the action space evolves during the submodular optimization process, which helps demonstrate how DynaAct progressively refines the candidate set to balance utility and diversity.
>
> Given the reasoning context:
>
> > Question: Is the Earth the center of the universe?
> >
> >
> > *Answer:* I clarify the problem by identifying the assumptions made in the past and restating them in my own words to better understand how they shaped the current situation. Early models placed Earth at the center, like Ptolemy’s.
> >
>
> The change in the action space across optimization steps is shown below:
>
> |  | Action Space | Submodular Score |
> | --- | --- | --- |
> | step 1 | **action 1**: I need to identify the underlying structure or logic behind the current understanding, and examine how it fits into a broader conceptual or analytical framework; **action 2**: I look for opportunities to reinterpret the problem using analogies or alternative viewpoints, questioning traditional interpretations and seeing if other models from my experience could offer a better fit. | 0.61 |
> | step 2 | **action 1**: I need to identify the underlying structure or logic behind the current understanding, and examine how it fits into a broader conceptual or analytical framework; **action 2**: I look for opportunities to reinterpret the problem using analogies or alternative viewpoints, questioning traditional interpretations and seeing if other models from my experience could offer a better fit; **action 3** (new): I clarify the problem by re-examining its foundational principles and constraints. | 0.87 |
> | step 3 | **action 1**: I need to identify the underlying structure or logic behind the current understanding, and examine how it fits into a broader conceptual or analytical framework; **action 2**: I look for opportunities to reinterpret the problem using analogies or alternative viewpoints, questioning traditional interpretations and seeing if other models from my experience could offer a better fit; **action 3**: I clarify the problem by re-examining its foundational principles and constraints; **action 4** (new): I clarify the problem by identifying the initial assumptions that might have shaped the current understanding. | 0.89 |
>
> This example highlights how the action set grows more diverse and focused with each step, improving the coverage of reasoning strategies relevant to the given context. We appreciate your suggestion and believe this qualitative illustration helps clarify the internal dynamics of our method and enhances its interpretability.
>
> ---
>
> ### **Q3: Comparison with few-shot prompting / dynamic exemplar selection**
>
> Thank you for drawing this insightful connection. To better contextualize our method, we compared DynaAct with strong few-shot prompting baselines, including GritLM, Gecko, and Instructor:
>
> | Method | MMLU | MMLUPro | GPQA | ARC-C | GSM8K | MATH-500 |
> | --- | --- | --- | --- | --- | --- | --- |
> | GritLM | 68.3 | 42.8 | 30.3 | 82.9 | 79.3 | 48.4 |
> | GritLM @sc16 | 69.4 | 46.2 | 35.4 | 85.2 | 85.3 | 51.4 |
> | Gecko | 68.8 | 43.2 | 29.8 | 82.1 | 78.0 | 46.2 |
> | Gecko @sc16 | 69.8 | 46.6 | 34.3 | 83.7 | 85.0 | 50.8 |
> | Instructor | 68.8 | 44.2 | 27.8 | 81.1 | 79.9 | 49.0 |
> | Instructor @sc16 | 68.9 | 47.3 | 34.8 | 84.9 | 85.7 | 50.6 |
> | **DynaAct** | **70.2** | **51.4** | **39.4** | **88.3** | **89.2** | **61.0** |
>
> These results show that DynaAct consistently outperforms competitive few-shot baselines across all tasks, reinforcing its effectiveness.
>
> ---
>
> ### **Q4: Release of observations from Section 4.4**
>
> Yes, absolutely. We are committed to open-sourcing the observations described in Section 4.4 to support reproducibility and foster further research in this area. We appreciate your encouragement on this front.
>
> ---
>
> Once again, thank you for your thoughtful and constructive feedback. We believe your suggestions have helped us further validate and clarify the contributions of our work.

---

> > ### Comment · Reviewer_ML8s · 2025-08-05
> > **Thank you for your rebuttal**
> >
> > Dear Reviewer,
> >
> > Thank you for the updated experiments and clarification, which have solved my major concerns. I have raised my score accordingly.

---

> > > ### Author Response · Authors · 2025-08-06
> > > **Sincere thanks for your feedback!**
> > >
> > > Thanks so much for reviewing our response and the updated experiments. We're relieved that addressed your main concerns, and we really appreciate you adjusting your score.

---

### Official Review · Reviewer_p9Ye · 2025-07-03

**Clarity:** 3
**Significance:** 3
**Originality:** 3
**Rating:** 5
**Confidence:** 3

**Summary:**

In this paper the authors introduce DYNAACT, a framework to construct compact action spaces dynamically to enhance reasoning in sequential decision-making tasks. Their method employs submodular optimization to balance utility and diversity in action selection, ensuring scalability and efficiency. The authors performed extensive experiments across six benchmarks (MMLU, GPQA, ARC-C, GSM8K, MATH-500) to demonstrate the strength of their approach. DYNAACT achieved 6.8% higher accuracy on MATH-500 compared to other baselines.

**Questions:**

Can this be applied to tasks with highly dynamic and unpredictable action spaces? For example in robotics?

Sub modular optimization can be computationally expensive. Do you see any scalability problems related to this for large complex domains?

**Ethical Concerns:**

["NO or VERY MINOR ethics concerns only"]

**Final Justification:**

The authors have addressed most of my concerns. Overall, I think it is a good paper.

**Limitations:**

Yes

**Quality:**

3

**Strengths And Weaknesses:**

The authors provided a rigorous methodology for dynamic action space construction using submodular functions, and through extensive experiments validated their approach across six benchmarks.

The paper is well-written, with clear explanations of the submodular optimization approach and experimental results.

The focus on enhancing reasoning in sequential decision-making using large language models is highly relevant. The paper has demonstrated the importance of well defined action spaces in MDP based reasoning. Their experimental analysis shows promise of this approach.

The dynamic action space construction is a novel research question, and the use of submodular functions to balance utility and diversity in action selection seems to be novel.

---

> ### Author Rebuttal · Authors · 2025-07-30
>
> Thank you very much for your thoughtful and constructive review. We truly appreciate your recognition of our work and your encouraging feedback on its clarity, novelty, and significance. We will also make sure to incorporate your insightful suggestions into the final version of the paper. Below, we address your specific questions in detail:
>
> ---
>
> ### **Q1: Can this be applied to tasks with highly dynamic and unpredictable action spaces? For example in robotics?**
>
> We value your insightful question about the relevance of our method in fields with extremely dynamic and unpredictable action environments, like robotics. To investigate this, we carried out further experiments on **Alfworld** [1], a recognized benchmark for embodied agents. Alfworld features six categories of domestic chores that differ in complexity and unpredictability. Following the official evaluation protocol, we use success rate as the evaluation metric. We evaluated our approach using the unseen test set composed of 134 instances, averaging the results over 10 runs to minimize statistical variance.
>
> Here are the results:
>
> | Method | Pick | Look | Clean | Heat | Cool | Pick Two | Avg. |
> | --- | --- | --- | --- | --- | --- | --- | --- |
> | Llama-3.1-8B-Instruct | 34.2 | 45.0 | 10.3 | 7.4 | 9.5 | 11.8 | 18.7 |
> | SC@maj16 | 47.1 | 52.2 | 24.5 | 19.6 | 21.9 | 32.4 | 31.8 |
> | RAP | 47.9 | 49.4 | 31.6 | 17.8 | 20.5 | 33.5 | 32.8 |
> | rStar | 37.5 | 42.2 | 28.7 | 23.5 | 17.1 | 35.9 | 30.1 |
> | **DynaAct (Ours)** | **59.6** | **57.8** | **42.3** | **34.3** | **29.5** | **41.2** | **43.6** |
>
> As you can see, DynaAct substantially outperforms all baselines across all task types. This suggests that your concern is well-addressed in practice: DynaAct indeed generalizes effectively to settings with dynamic and complex action spaces, such as those found in robotics.
>
> ---
>
> ### **Q2: Submodular optimization can be computationally expensive. Do you see any scalability problems related to this for large complex domains?**
>
> Thank you for bringing up this significant matter. We recognize the worries about the computational expense of submodular optimization. To specify, according to our profiling and quantitative analysis (refer to Table 8), **the primary source of computational burden in our framework originates from the MCTS** component instead of the submodular optimization phase. Indeed, the submodular optimization we use is executed through a greedy algorithm with linear time complexity, which proves to be highly effective in real-world applications. Additionally, as mentioned in Appendix F.6, the main factor for scalability is the magnitude of the proxy action space. In our approach, DynaAct performs effectively even with extensive proxy action spaces owing to two key strategies: (1) **Storing action embeddings,** which prevents unnecessary calculations across various evaluations; and (2) **Linear-time greedy submodular selection,** which guarantees that potential actions can be chosen efficiently without exhaustive searching. Therefore, we are confident DynaAct can manage large and intricate domains without major computational limitations because of the submodular optimization itself.
>
> ---
>
> Thank you again for your insightful feedback. We trust that these explanations alleviate your concerns completely and showcase the broad applicability and effectiveness of our method.
>
> ---
>
> **Reference:**
>
> [1] Shridhar, M., Yuan, X., Côté, M. A., Bisk, Y., Trischler, A., & Hausknecht, M. (2020). Alfworld: Aligning text and embodied environments for interactive learning.

---

> > ### Comment · Reviewer_p9Ye · 2025-08-05
> > **Thanks for the new results**
> >
> > Thank you for answering my questions and also for the new results. I have updated my review.

---

> > > ### Author Response · Authors · 2025-08-06
> > > **Sincere thanks for your feedback!**
> > >
> > > Really appreciate you giving our new results a look - glad we could answer your questions properly. Thanks for updating your review based on what we provided.

---

### Official Review · Reviewer_eEVj · 2025-07-03

**Clarity:** 3
**Significance:** 3
**Originality:** 3
**Rating:** 4
**Confidence:** 4

**Summary:**

The authors develop a novel framework named DYNAACT for automatically constructing a compact action space to enhance sequential reasoning in complex problem-solving scenarios, which firstly estimates a proxy for the complete action space by extracting general sketches observed in a corpus covering diverse complex reasoning problems using large language models

**Questions:**

The core idea is to construct high-level actions (referred to as action subsets in the paper) to group similar low-level actions — an approach that is not novel in the LLM community.

I have a few questions and suggestions regarding your method:

1. How is the number of action subsets determined in your approach? Additionally, how does this number impact the overall performance of your method?
2. In my experience, embedding-based clustering tends to be fragile and often lacks interpretability, particularly in understanding the semantics of cluster centers. Have you considered using LLM-based clustering, where the categories of low-level actions are derived via LLM reasoning?
3. Llama3 series and some benchmarks, e.g. GSM8K, MATH500, I suggest evaluating your method on AIME24 and AIME25 using the Qwen3 model series to make the experimental results more convincing and up-to-date.

**Ethical Concerns:**

["NO or VERY MINOR ethics concerns only"]

**Final Justification:**

Provide new results of Qwen3 series on AIME24 and AIME25, and the feedback address most of my concerns.

**Quality:**

3

**Strengths And Weaknesses:**

1. the paper is well written and easy to follow
2. the technique details are sound
3. the novelty is not enough

---

> ### Author Rebuttal · Authors · 2025-07-30
>
> We truly appreciate your considerate and insightful feedback. We are pleased that you consider our paper to be well-written and technically sound. We meticulously respond to all of your questions and suggestions in depth below. We will also make sure to incorporate your insightful suggestions into the final version of the paper.
>
> ---
>
> ### **Q1: Novelty of high-level actions**
>
> We understand your concern that constructing high-level actions to group similar low-level actions has been previously explored in the literature, as noted in Section 1 of our manuscript. However, we would like to clarify that the key novelty of our work lies not in the general concept of high-level actions per se, but in the **principled construction** of a compact and scalable action space. We believe this direction — making high-level actions *scalable, compact, and generalizable* in a unified framework — addresses an under-explored and practically important challenge in the field.
>
> ---
>
> ### **Q2: How is the number of high-level actions determined?**
>
> I appreciate you bringing this to attention. To clarify, our framework does not involve partitioning actions into disjoint “subsets.” Rather, we dynamically construct a set of high-level actions to serve as the compact action space at each step of reasoning.
>
> In our current implementation, we empirically fix the number of high-level actions $m$ to 5, which strikes a good balance between expressiveness and search efficiency. We also conducted ablation studies to investigate how varying $m$ impacts performance. The results are summarized below:
>
> |  | MMLU | MMLUPro | GPQA | ARC-C | GSM8K | MATH-500 |
> | --- | --- | --- | --- | --- | --- | --- |
> | m = 3 | 69.3 | 50.1 | 37.9 | 88.1 | 86.3 | 57.6 |
> | m = 5 | 70.2 | 51.4 | 39.4 | 88.3 | 89.2 | 61.0 |
> | m = 7 | 70.3 | 51.9 | 38.9 | 89.1 | 87.3 | 60.2 |
> | m = 9 | 70.8 | 50.6 | 38.4 | 88.2 | 88.9 | 59.4 |
>
> As shown, $m=5$ achieves consistently strong performance across benchmarks, and increasing $m$ beyond this point provides diminishing or even negative returns. We will make this clearer in our paper revision.
>
> ---
>
> ### **Q3: Fragility of embedding-based clustering**
>
> You raised an excellent point about the limitations of embedding-based clustering. We agree that naive clustering of embeddings can be fragile. Indeed, our own experiments (see Appendix F.5) show that using pure embedding-based methods without additional regularization leads to unstable results.
>
> To address this, our method integrates **submodular selection as a principled approach** to ensure both coverage and diversity of action candidates. We also compared this with a variant where action categories are directly derived from LLM reasoning. Specifically, we use Llama-3.1-8B-Instruct to generate high-level actions at each step, conditioned on the trajectory so far, without applying submodular selection (**referred to as “– submodular” in Table 2**). The performance of this LLM-based variant was significantly lower due to reduced compactness and increased redundancy, which negatively impacts the MCTS exploration efficiency.
>
> Therefore, while we appreciate the benefit of pure LLM-driven clustering, our empirical findings suggest that submodular selection yields a better trade-off between semantic relevance and decision efficiency.
>
> ---
>
> ### **Q4: Evaluation on Qwen3 and AIME benchmarks**
>
> Thank you for suggesting additional evaluation on the Qwen3 model series and AIME24/25 benchmarks. We fully agree that incorporating newer models and more challenging datasets can strengthen the empirical foundation of our work.
>
> We would like to clarify that the Qwen3 series was released only about two weeks before the submission deadline, which limited our ability to include these models in our original experiments. Nonetheless, we have since conducted a comprehensive set of evaluations using **Qwen3** Series on **AIME 2024** and **AIME 2025**
>
> Below are the results:
>
> ### **Qwen3-1.7B**
>
> | Method | AIME 2024 | AIME 2025 |
> | --- | --- | --- |
> | Qwen3-1.7B | 13.4 | 9.8 |
> | SC@maj16 | 23.3 | 13.3 |
> | RAP | 23.3 | 16.7 |
> | rStar | 20.0 | 13.3 |
> | **DynaAct** | **30.0** | **23.3** |
>
> ### **Qwen3-4B**
>
> | Method | AIME 2024 | AIME 2025 |
> | --- | --- | --- |
> | Qwen3-4B | 25.0 | 19.1 |
> | SC@maj16 | 33.3 | 20.0 |
> | RAP | 30.0 | 23.3 |
> | rStar | 30.0 | 20.0 |
> | **DynaAct** | **36.7** | **26.7** |
>
> ### **Qwen3-8B**
>
> | Method | AIME 2024 | AIME 2025 |
> | --- | --- | --- |
> | Qwen3-8B | 29.1 | 20.9 |
> | SC@maj16 | 33.3 | 20.0 |
> | RAP | 30.0 | 23.3 |
> | rStar | 36.7 | 23.3 |
> | **DynaAct** | **46.7** | **30.0** |
>
> ### **Qwen3-14B**
>
> | Method | AIME 2024 | AIME 2025 |
> | --- | --- | --- |
> | Qwen3-14B | 31.7 | 23.3 |
> | SC@maj16 | 36.7 | 23.3 |
> | RAP | 33.3 | 26.7 |
> | rStar | 36.7 | 20.0 |
> | **DynaAct** | **43.3** | **33.3** |
>
> ### **Qwen3-32B**
>
> | Method | AIME 2024 | AIME 2025 |
> | --- | --- | --- |
> | Qwen3-32B | 31.0 | 20.2 |
> | SC@maj16 | 36.7 | 26.7 |
> | RAP | 33.3 | 23.3 |
> | rStar | 36.7 | 23.3 |
> | **DynaAct** | **46.7** | **30.0** |
>
> These results clearly demonstrate that **DynaAct consistently improves the problem-solving capabilities of Qwen3 models**, often by large margins over the strong baselines.
>
> We hope these additional results effectively address your concerns and further strengthen the empirical impact of our work.
>
> ---
>
> We appreciate your thorough review and valuable comments once more. Your feedback has improved our understanding of how to convey the value and merits of our work, and we sincerely value your suggestions.

---

> > ### Comment · Reviewer_eEVj · 2025-08-03
> >
> > Thanks for your new results of Qwen3 series on AIME24 and AIME25, and the feedback address most of my concerns. I will raise my score.

---

> > > ### Author Response · Authors · 2025-08-03
> > > **Sincere thanks for your feedback!**
> > >
> > > Thanks for taking the time to look over our work - your feedback was really helpful. We're glad our clarifications cleared things up and appreciate you taking another look at it. We'll definitely incorporate those additional details you mentioned to strengthen the manuscript.

---

### Decision · Program_Chairs · 2025-09-17

**Decision:**

Accept (poster)

**Comment:**

This paper identifies the limitation of the current framework on MDP-style LLM reasoning on lacking a scalable and compact action space. In response, the authors propose to first mine action space from existing corpora and formulate a submodular function to evaluate the utility and diversity of the actions to form a dynamic action space. Extensive experiments on six tasks show good performance improvement over previous methods with good latency control.

The paper addresses an important problem in current MDP-style reasoning methods in LLMs. The proposed method is relatively novel and backed by experiments on various backbones and benchmarks both in Q&A and maths (considering results in the paper and added during discussion period as requested by reviewers). In addition, application to tasks with highly dynamic and unpredictable action spaces was discussed during rebuttal phase and importance of using submodular selection vs embedding based methods were emphasized. The paper would become stronger by adding  more qualitative results.

I ask the authors to please include the results and important clarifications discussed during discussion period to the camera ready version of the paper.